# Performance Enhancement of Silver Nanowire-Based Transparent Electrodes by Ultraviolet Irradiation

**DOI:** 10.3390/nano12172956

**Published:** 2022-08-26

**Authors:** Shengyong Wang, Huan Liu, Yongqiang Pan, Fei Xie, Yan Zhang, Jijie Zhao, Shuai Wen, Fei Gao

**Affiliations:** School of Optoelectronic Engineering, Xi’an Technological University, Xi’an 710021, China

**Keywords:** AgNWs, ultraviolet, welding, transparent electrodes

## Abstract

Silver nanowires (AgNWs) are used as transparent electrodes (TE) in many devices. However, the contact mode between the nanowires is the biggest reason why the sheet resistance of silver nanowires is limited. Here, simple and effective ultraviolet (UV) irradiation welding is chosen to solve this problem. The influence of the power density of the UV irradiation on welding of the silver nanowires is studied and the fixed irradiation time is chosen as one minute. The range of the UV (380 nm) irradiation power is chosen from 30 mW/cm^2^ to 150 mW/cm^2^. First of all, the transmittance of the silver nanowire film is not found to be affected by the UV welding (400–11,000 nm). The sheet resistance of the silver nanowires decreases to 73.9% at 60 mW/cm^2^ and increases to 127.6% at 120 mW/cm^2^. The investigations on the UV irradiation time reveal that the sheet resistance of the AgNWs decreases continuously when the UV irradiation time is varied from 0 to 3 min, and drops to 57.3% of the initial value at 3 min. From 3–6 min of the continuous irradiation time, the change of the sheet resistance is not obvious, which reflects the self-limiting and self-termination of AgNWs welding. By changing the wavelength of the UV irradiation from 350–400 nm, it is found that the welding effect is best when the UV wavelength is 380 nm. The average transmittance, square resistance, and the figure of merit of the welded AgNWs at 400–780 nm are 95.98%, 56.5 Ω/sq, and 117.42 × 10^−4^ Ω^−1^, respectively. The UV-welded AgNWs are also used in silicon-based photodetectors, and the quantum efficiency of the device is improved obviously.

## 1. Introduction

The rapid development in flexible transparent electrodes (TE) has opened up possibilities for using them in a wide range of applications, such as optoelectronic devices. At present, many devices still use indium tin oxide (ITO) as a transparent electrode. This is because ITO has excellent conductivity (10–20 Ω/sq) and high transparency (transmittance of 90%) [1]. However, ITO is brittle, and its constituent raw materials are scarce. These factors prevent ITO from being widely used in future flexible optoelectronic devices [2,3,4,5]. Overall, this impacts the development of flexible electronic devices.

In order to overcome these limitations, a number of alternative transparent electrode materials have been suggested for replacing ITO, including carbon nanotubes (CNTs) [6], graphene [7], metal grids [8], copper and silver metal nanowires [9,10], and electrospun metal nanofibers. Among the transparent conductive films (TCFs), metal nanowires, especially AgNWs, exhibit superior performance, including low sheet resistance (10–20 Ω/sq), and high transparency (80–90%) [11,12,13]. Moreover, it can also be applied to flexible devices, which widens its application range greatly. However, AgNW-based TCFs exhibit some drawbacks including high surface roughness, high resistance between the AgNW junctions, and weak adhesion to various substrates. Furthermore, AgNW-based TCFs are easily oxidized when they are exposed to water or air, which results in a sharp increase in the sheet resistance [14,15].

In recent years, the optimization of AgNWs has been widely studied. The sheet resistance of AgNWs depends on the close connection at the junction [16], and the contact resistance at the junction strongly affects the total sheet resistance of AgNWs electrodes. Through welding, the sheet resistance and the roughness of the film can be reduced without reducing the transmittance of the AgNWs. Many welding methods have been proposed, such as thermal welding [17], plasmonic welding [18,19], chemical welding [20,21,22], and UV irradiation welding [23]. Among these, the UV irradiation welding does not cause other effects on the film compared to the other welding methods. It also does not produce a large amount of heat to damage the substrate. The UV irradiation welding, therefore, has a better potential of improving the properties of AgNWs films. Liang et al. studied the efficient welding of AgNWs by using ultraviolet A (UVA) nano-photothermal process with A wavelength range of ~320 to ~400 nm [23]. They obtained AgNWs with a low sheet resistance of 25 Ω/sq, a high transmittance of 90%, and exhibiting excellent flexibility. KOU et al. found that the AgNWs network still achieved sheet resistance of less than 20 Ω/sq, transmittance of ~87% at 550 nm wavelength, and good mechanical flexibility after 1 h or more of sunlight exposure [24]. During the UV welding, the light-induced welding was a self-limiting process due to the gradually weakened plasmonic effect during the formation of nanowire joints. However, the current experimental light sources mostly have a fixed optical power and a broad spectral range. Therefore, there is a lack of systematic research on the optical power density and irradiation wavelength of UV irradiation.

In this work, the light power irradiated on the film was changed with a fixed irradiation time of 1 min. It was found that when the UV power irradiated on the AgNWs was too high, the resulting sheet resistance was also high. At the same time, the silver nanowires were irradiated by the UV light at 60 mW/cm^2^ for 0–6 min. After the irradiation for 1 min, the initial welding of the AgNWs occurred, and the wire-to-wire contact changed significantly as compared to that before the welding. After 3 min of the irradiation, the welding was basically over, and the sheet resistance did not change significantly. Under the condition of changing the wavelength of the UV light, the most suitable UV wavelength was determined by comparing the change of sheet resistance. Finally, the soldered AgNWs were applied to silicon-based photodetectors.

## 2. Materials and Methods

### 2.1. Preparation of the AgNWs

AgNWs with diameters of 30 nm were purchased from KECHUANG. In this experiment, 2.0 mg/mL AgNWs were selected. In order to avoid an uneven dispersion of the AgNWs, the reagent had to be placed on the vortex mixer and was made to vibrate for 1 min before the spin coating. The AgNWs were then spread all over the BaF_2_ substrate with a pipette. In order to evenly distribute the solution on the substrate, the homogenizer should be started after the solution on the substrate is stationary for 10 s. The speed of the homogenizer is 1500 r/min, and the time is 1 min.

### 2.2. UV Irradiation Welding of AgNWs

The UV irradiation process is shown in Figure 1. The AgNWs film coated on BaF_2_ substrate was irradiated by a Microenerg 500 W UV light source. The distance between the sample and the UV light source was adjusted by a lifting platform to control the light power density of the UV lamp. As can be seen in Figure 1, the overlapping regions of the silver nanowires will become more tightly connected when exposed to UV radiation. The wavelength of the UV lamp was controlled by the filter installed in the light outlet. 

### 2.3. Characterization

The optical transmittance spectra were obtained by using a spectrophotometer (Lambda 950, Perkin Elmer, MA, USA) and Fourier infrared spectrometer (VERTEX 70V, BRUKER, Saarbrücken, Germany). The sheet resistance and the electrical properties of the films were measured by a four-probe square resistance tester (ST2558B-F01, SUZHOU JINGGE, Suzhou, China). Field emission scanning electron microscopy (SEM, Zeiss, Sigma300, Oberkochen, Germany) was also used to investigate the microstructure of thin films. The optical power shining on the sample was measured by an optical power meter (Thorlabs, PM160, Newton, MA, USA).

## 3. Results and Discussion

### 3.1. Optical Properties

It is well known that the two absorption peaks of AgNWs in the UV-visible range generally appear at about 350 and 380 nm, which is derived from the plasmon resonance of AgNWs, similar to bulk silver film and the transverse plasmon mode of AgNWs, respectively [25,26]. Optical properties are one of the most important properties of transparent conductive films. Figure 2 shows the transmittance curve of AgNWs on the BaF_2_ substrate after 0–6 min of the UV irradiation. 

As can be seen from the figure, the transmittance of AgNW film has not been affected significantly after the UV irradiation (380 nm, 60 mW/cm^2^) at different irradiation times. It also does not exhibit any appreciable change compared to that before the irradiation. When AgNWs network is irradiated by the UV light, surface plasmon resonance is generated, which is a photothermal effect on AgNWs network. This increases the local temperature of AgNWs network and forms a so-called “hot spot” [27]. Therefore, the way the UV light is irradiated on the sample does not affect the transmittance of AgNWs. The results presented in Figure 2 also prove this phenomenon. With the change of irradiation time, the transmittance fluctuates only slightly irregularly, which may be caused by the subtle differences between the samples coated with spins and the substrate preparation process. At the same time, when the influence of other factors of the UV welding on the transmittance of AgNWs are investigated, the same conclusions are drawn.

### 3.2. Electrical Properties

The most valuable benefit of the UV irradiation on AgNWs is to improve their electrical properties. The effects of the UV irradiation with different optical power densities on the properties of AgNWs are studied. The irradiation time is controlled at 1 min because the influence of the first minute of the irradiation on the AgNW film can be readily observed. The optical power density of the UV light varies from 30 to 150 mW/cm^2^. In order to avoid the error of sheet resistance affecting the experimental law, the relative sheet resistance value is adopted. It can be seen from Figure 3a (Rs/Rs_0_, Rs: real-time sheet resistance, Rs_0_: initial sheet resistance) that the Rs/Rs_0_ decreases to 0.70484 when the UV power density is 60 mW/cm^2^, and increases to 1.56775 when the UV power density is 150 mW/cm^2^. The increase in the sheet resistance can be attributed to the high power of the UV light accelerating the precipitation of degradation particles on the surface of the AgNWs. Accelerated oxidation and vulcanization by the UV light result in the presence of small nanoparticles and thus increase the sheet resistance [28]. When studying the effect of UV irradiation time on silver nanowire film, the UV power density is fixed as 60 mW/cm^2^. As can be seen from Figure 3b (380 nm, 60 mW/cm^2^), the change of the sheet resistance of the AgNWs is obvious in the first 3 min, and Rs/Rs_0_ reaches 0.56596 in the third minute. It can also be seen in Figure 3b that 3 min after the welding, the AgNW sheet resistance does not decrease further. This is due to the fact that the welding process of AgNWs has been concluded. This implies that the UV irradiation welding of AgNWs is a self-limiting process [14]. This characteristic, as compared to the heat-welding method [17] has an obvious advantage. It can be seen from Figure 3c that the fixed irradiation power is 60 mW/cm^2^ and the time is 3 min. When the UV wavelength is 380 nm, Rs/Rs_0_ is the lowest, which is the same as the UV absorption peak of AgNWs. This phenomenon also proves that the UV welding of the AgNWs depends on the absorption of the UV light. Table 1 shows the variation of square resistance of silver nanowires before and after welding under the influence of different factors.

In Figure 3a, when the UV welding light power is too low, the influence on the film is relatively small, and the error bar is very small. However, with the gradual increase in the optical power, the welding effect and particle precipitation become more obvious, which also leads to the increase in the error bar. As shown in Figure 3b, with the extension of welding time, the error bar is large in the second minute because the welding is not completely completed, and the error bar is small when the welding is basically completed three minutes later. In Figure 3c, the error bar of the test results near the UV irradiation wavelength of 380 nm is small. This is caused by the strong absorption capacity of silver nanowires to 380 nm UV light, resulting in better welding results.

### 3.3. Figure of Merit

The transmittance and sheet resistance of transparent electrodes is of high importance for their application. The film transmittance and sheet resistance define the figure of merit of a transparent electrode. Haacke equation [29] is used for the calculation of the figure of merit, which is given as:(1)$$\Phi_{TC}=\frac{T^{10}}{R_{sh}}$$
where Rsh is the average sheet resistance and *T* is the optical transmittance (here, transmittance has been averaged over the wavelength range of 400–11,000 nm). The *Φ_TC_* values of AgNWs films are shown in Figure 4 [30].

The results of the figure of merit calculation also confirm the previous conclusions. According to these results, it can be concluded that the best welding effect is obtained by selecting the optical power density of 60 mW/cm^2^, the irradiation time of 3 min, and the irradiation wavelength of 380 nm.

### 3.4. Microstructural Properties

SEM images give an insight into the welding process of AgNWs. Figure 5a–c show SEM images of the AgNWs welded for 0 min, 1 min, and 3 min, respectively. The areas highlighted by the black circles are where the silver nanowires are welded. Before the UV irradiation, the connection mode between the AgNWs is mostly lap connection, which is an important factor affecting its electrical properties. After 1 min of the UV irradiation, it can be seen that the AgNWs have changed from the lap to preliminary welding. At 3 min, the welding of the AgNWs is basically finished, and the welding degree is significantly higher than that observed for the irradiation time of 1 min. By comparing the effects of different UV irradiation times on the microstructure of AgNWs, it can be concluded that the UV irradiation plays a key role in the welding of the AgNWs. 

Figure 5d–f are the inclined plane SEM images that were obtained for observing the precipitation of small particles of AgNWs. The surface of the AgNWs without the UV irradiation is smooth, and no small particles appear on the surface. When comparing the surface of AgNWs before and after the welding, it can be clearly seen that the precipitation of small nanoparticles on the surface increases significantly after the UV irradiation. This also indicates that the welding optimization and degradation effect of the UV irradiation on AgNWs occur simultaneously [28]. Figure 6 is a supplement to the previous microscopic morphological representation. Figure 6a mainly shows the welding effect of silver nanowires in many places, and Figure 6b mainly shows the phenomenon of particle precipitation after ultraviolet light irradiation.

### 3.5. The Device Application

The AgNWs welding technology is applied to simple silicon devices [31]. The structure of the device is Ag/Si/AgNWs/Ag, in which the upper and lower silver are ring electrodes for eliciting test signals, the silver nanowire is coated on the upper surface of the whole silicon as a transparent electrode, and silicon is the active layer of the whole device. AgNW-contained solution is spin-coated on a Si substrate, which spontaneously forms a Schottky device. Compared with unwelded silver nanowires, due to the good transparency of the AgNWS conductive layer, more photons can be driven into the photoactive Si material, thus enhancing the light response. However, the contact between the welded silver nanowires and Si substrate is closer, and it provides more ways to transmit electrons. UV welding reduces the sheet resistance of the Ag nanowires. UV light is used to treat AgNWs coated on silicon, and a comparison of the device performance for two methods is made. It can be seen from Figure 7 that the quantum efficiency at 550 nm increases from 36.1% to 57.5% before and after the UV welding. The sheet resistance of the welded AgNWs decreases and more carriers are injected into the organic layer through the anode, thus improving the quantum efficiency of the device [32,33].

Since the transparent electrode of silver nanowire is generally located above the whole device, UV irradiation will affect the whole device. The effect is not too great when the photosensitive layer is more stable silicon, but should be discussed when UV sensitive materials act as photosensitive layers. For example, ZnO-based inverted quantum dot light-emitting diode (QLED) can improve the carrier injection of QLED after UV irradiation, which causes the device to exhibit high quantum efficiency [34]. However, some studies have shown that the perovskite materials in the perovskite base will decompose after UV irradiation, resulting in the reduction of the photostability of the device [35,36]. Therefore, before using UV irradiation welding, it is necessary to determine whether there is an effect on other layers of the device.

Based on the device performance comparison, it can be concluded that the UV welding can improve the performance of the device, which provides help for the fabrication of the device in the future. It should be noted here that the quantum efficiency value of the standard silicon is the standard value in this test system (the device structure and type of standard silicon are different from this device).

## 4. Conclusions

Influences of the UV irradiation power, time, and wavelength on the welding of AgNWs were studied. The welding of the AgNWs was realized based on the nanoscale photothermal process. Under the UVA irradiation, the sheet resistance of the AgNW films decreased rapidly within 3 min without affecting the optical transmittance. When the average transmittance of the AgNW films was 85.2% at 400–2000 nm and 76.6% at 2000–10,000 nm, the sheet resistance of the AgNW films decreased to 57.3%. Overall, an improvement in the photoelectric performance of the films was observed when the UV power was gradually increased. It was also observed that when the UV power is too high, the precipitation of small particles on the surface of the AgNWs was accelerated, thus improving the sheet resistance of the AgNWs. The welding effect was best when the UV wavelength was 380 nm, which was consistent with the absorption peak of the AgNWs. Finally, this method (380 nm, 60 mW/cm^2^, 3 min) was applied to silicon devices, and the performance was found to be significantly improved compared with those without the UV irradiation.

## Figures and Tables

**Figure 1 nanomaterials-12-02956-f001:**
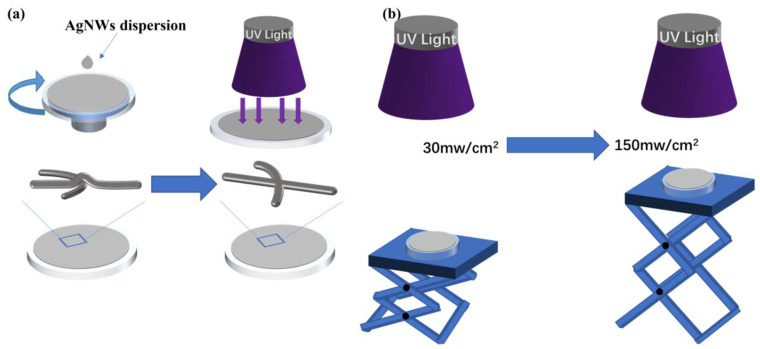
Schematic diagram of the UV irradiation of AgNWs. (**a**) Schematic diagram of welding principle. (**b**) Schematic diagram of optical power regulation principle.

**Figure 2 nanomaterials-12-02956-f002:**
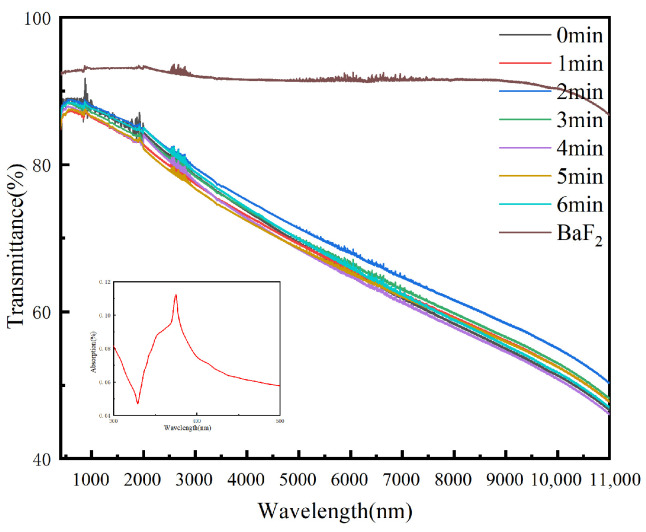
Transmittance of AgNWs with different UV irradiation times (400–11,000 nm). Insert of the figure shows the absorption of AgNWs at 300–500 nm.

**Figure 3 nanomaterials-12-02956-f003:**
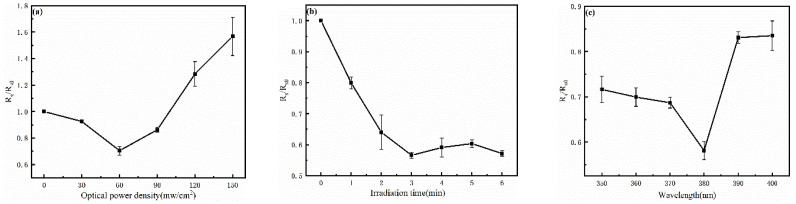
Influence of different factors on the sheet resistance of the UV welded AgNWs. (**a**) Sheet resistance of AgNWs varies with the UV irradiation power. (**b**) Sheet resistance of AgNWs varies with the UV irradiation time. (**c**) Sheet resistance of AgNWs varies with the UV wavelength.

**Figure 4 nanomaterials-12-02956-f004:**
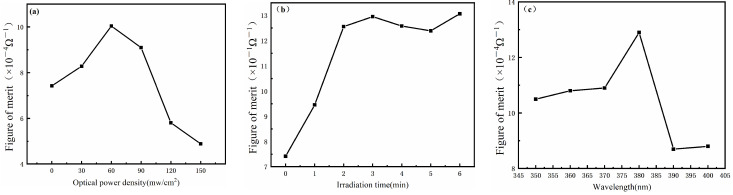
Figure of merit diagram showing the influence of different factors on UV welding. (**a**) Optical power density (**b**) Irradiation time (**c**) Ultraviolet wavelength.

**Figure 5 nanomaterials-12-02956-f005:**
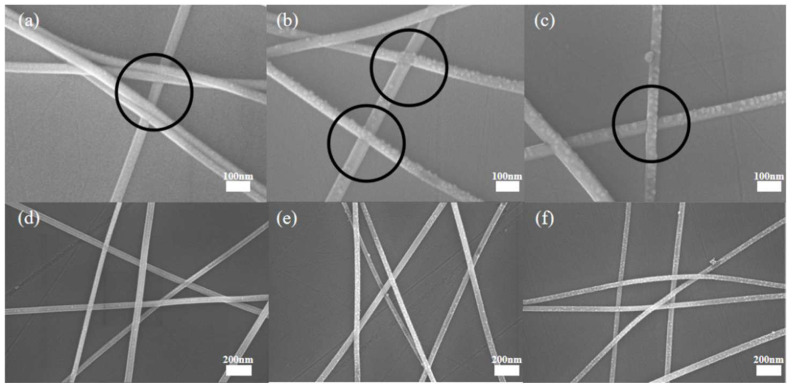
SEM image of AgNWs: (**a**–**c**) are oblique (30°) SEM images to observe the overlapping of AgNWs, (**d**–**f**) are the inclined plane SEM images to observe the precipitation of AgNW small particles. (**a**,**d**) have UV irradiation for 0 min, (**b**,**e**) for 1 min, and (**c**,**f**) for 3 min.

**Figure 6 nanomaterials-12-02956-f006:**
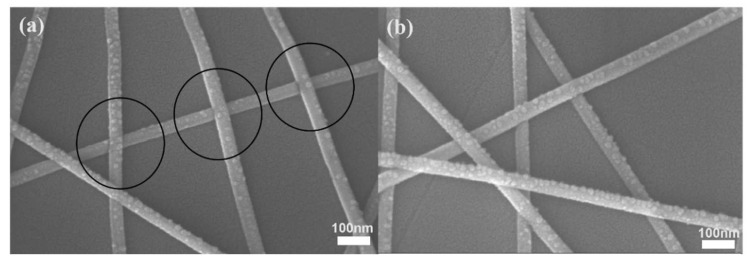
SEM images of AgNWs irradiated by the UV light. (**a**) AgNW welding. (**b**) AgNWs precipitate small particles.

**Figure 7 nanomaterials-12-02956-f007:**
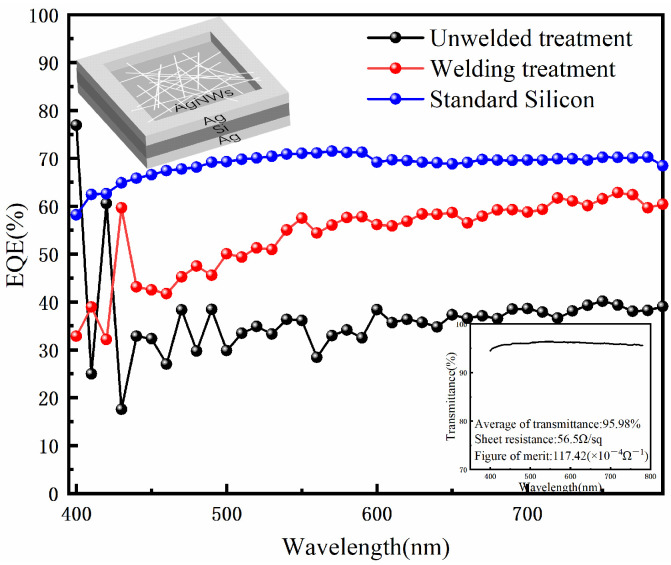
Comparison of quantum efficiency of silicon devices before and after the UV welding. Insert of the figure shows the transmittance of AgNWs at 400–780 nm as well as the values of the average transmittance, the sheet resistance, the figure of merit after the irradiation, and the device structure diagram.

**Table 1 nanomaterials-12-02956-t001:** Variation of sheet resistance of AgNWs under ultraviolet irradiation under different influencing factors.

**Influence Factor**		**30 mw**	**60 mw**	**90 mw**	**120 mw**	**150 mw**	**-**
Optical power density	Before irradiation(R_s0_)/Ω·sq^−1^	98.5928	114.323	107.0665	101.30134	112.1804	-
	After irradiation(R_s_)/Ω·sq^−1^	88.369775	84.4353	87.34624	129.2432	170.1116	-
	R_s_/R_S0_	0.8963	0.7386	0.8158	1.2758	1.5164	-
**Influence Factor**		**1 min**	**2 min**	**3 min**	**4 min**	**5 min**	**6 min**
Irradiation time	Before irradiation(R_s0_)/Ω·sq^−1^	114.323	126.213	128.354	122.7258	106.1785	124.06775
	After irradiation(R_s_)/Ω·sq^−1^	89.77218	77.87644	73.54312	72.39732	69.75925	72.6567
	R_s_/R_S0_	0.79905	0.64003	0.56596	0.59091	0.6036	0.57137
**Influence Factor**		**350 nm**	**360 nm**	**370 nm**	**380 nm**	**390 nm**	**400 nm**
Ultraviolet wavelength	Before irradiation(R_s0_)/Ω·sq^−1^	123.2068	114.678	113.164	128.354	109.01	117.949
	After irradiation(R_s_)/Ω·sq^−1^	88.8007	81.857	77.8225	73.54312	92.6678	99.147
	R_s_/R_S0_	0.7207	0.7138	0.6877	0.57297	0.85	0.8406

## Data Availability

The data supporting the findings of this study are available by reasonable request to wangshengyong@st.xatu.edu.cn.

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
