# Peer review of "Performance Enhancement of Silver Nanowire-Based Transparent Electrodes by Ultraviolet Irradiation"

_nanomaterials, 2022, doi:10.3390/nano12172956_

Round 1

Reviewer 1 Report

At the moment, flexible transparent electrodes attract a significant interest. To use the advantages of these structures widely, contacts play an important role. The manuscript describes how to improve the contact properties of Ag nanowires using quite simple but effective procedure.  Main parameters were determined to optimize electrical properties of these contacts. There are just few points which have to be discussed in more detail.

1. The device application part (3.5) has to be extended. A device structure must be presented in order to understand the mechanism of IQE improving "The sheet resistance of the welded AgNWs decreases and more carriers are injected into the organic layer through the anode, thus improving the quantum efficiency of the device" is not enough.

2. The influence of the UV irradiation on more delicate devices has to be discussed.

After clarification of these points the paper can be published in the journal.

Author Response

1.The device application part (3.5) has to be extended. A device structure must be presented in order to understand the mechanism of IQE improving "The sheet resistance of the welded AgNWs decreases and more carriers are injected into the organic layer through the anode, thus improving the quantum efficiency of the device" is not enough.

Thank you for the reviewer’s suggestion for this paper, more experimental details have been added.

The structure of the device is Ag/Si/AgNWs/Ag, in which the upper and lower silver are ring electrodes for eliciting test signals, and the silver nanowire is coated on the upper surface of the whole silicon as a transparent electrode, and silicon is the active layer of the whole device. AgNW-contained solution was spin-coated on a Si substrate, which spontaneously forms a Schottky device[33]. Compared with uncoated silver nanowires, due to the good transparency of the AgNWS conductive layer, more photons can be driven into the photoactive Si material, thus enhancing the light response. Meanwhile, the contact between the welded silver nanowires and Si substrate is closer, which also reduces the sheet resistance of the silver nanowires, and more charge carriers are injected into the organic layer through the anode, thus improving the quantum efficiency of the device. 

[33] Kim, H.-S.; Patel, M.; Kim, H.; Kim, J.-Y.; Kwon, M.-K.; Kim, J. Solution-Processed Transparent Conducting Ag Nanowires Layer for Photoelectric Device Applications. Materials Letters 2015, 160, 305–308, doi:10.1016/j.matlet.2015.07.142.

2. The influence of the UV irradiation on more delicate devices has to be discussed.After clarification of these points the paper can be published in the journal.

Based on the reviewer’s suggestion, The effect of UV irradiation on more delicate devices has been discussed in the paper.

Since the transparent electrode of silver nanowire is generally located above the whole device, UV irradiation will affect the whole device. The effect is not too great when the photosensitive layer is more stable silicon, but should be discussed when UV sensitive materials act as photosensitive layers. For example, ZnO based inverted quantum dot light-emitting diode (QLED) can improve the carrier injection of QLED after UV irradiation, which makes the device exhibit high quantum efficiency[34]. However, some studies have shown that the perovskite materials in the perovskite base will decompose after UV irradiation, resulting in the reduction of the photostability of the device[35,36]. Therefore, before using UV irradiation welding, it is necessary to determine whether there is an effect on other layers of the device

[34] Luo, Y.; Wang, J.; Wang, P.; Mai, C.; Wang, J.; Yap, B.K.; Peng, J. Effects of UV Irradiation and Storage on the Performance of Inverted Red Quantum-Dot Light-Emitting Diodes. Nanomaterials 2021, 11, 1606, doi:10.3390/nano11061606.

[35] Aristidou, N.; Sanchez-Molina, I.; Chotchuangchutchaval, T.; Brown, M.; Martinez, L.; Rath, T.; Haque, S.A. The Role of Oxygen in the Degradation of Methylammonium Lead Trihalide Perovskite Photoactive Layers. Angew. Chem. Int. Ed. 2015, 54, 8208–8212, doi:10.1002/anie.201503153.

[36] Nan, G.; Zhang, X.; Lu, G. Self-Healing of Photocurrent Degradation in Perovskite Solar Cells: The Role of Defect-Trapped Excitons. J. Phys. Chem. Lett. 2019, 10, 7774–7780, doi:10.1021/acs.jpclett.9b03413.

Reviewer 2 Report

This manuscript by Wang and co-workers reports a work about the influence of ultraviolet (UV) irradiation on the performance enhancement of silver nanowire (AgNW) based transparent electrodes upon transmittance, sheet resistance and the application on silicon photodetector. It is an interesting topic related to transparent electrodes upon their application. The result is interesting but inconclusive and short of novelty. It will be more suitable to publish in a special journal rather than Nanomaterials.

Here are only some comments:

1.  The authors focus on the interplay between UV light density, wavelength, and irradiation time on AgNWs films as a three-dimensional picture. However, there are only few points used to  conclude the results. The most data are only related to 380 nm UV light within  6 mins illumination.  Different wavelengths may need variable illumination time or density to reach optimal effects, but no data are presented.

2.  The whole manuscript only shows the relative sheet resistance, it will nor mean too much, it just indicates the influence of the UV light. The absolute sheet resistance value upon different illumination parameters is much more important than the relative one.

     3.  Figure 1 is only a sketch of the experiment, how to conclude “As can be seen from Figure 1, the AgNWs changed from lap to tighter welding state, which was also the main reason for the improvement of the electrical performance of the AgNWs”?

     4.  More precise description of the experiment is needed. “Subsequently, for a period of 10 s, the samples were put at the homogenizer for 1 min, which operated at the speed of 1500 rpm”.

      5.  Figure 2 should start from e.g., 350 nm, the most absorption feature of AgNWs should be included.  Why the transmittance at 2000 nm in the two parts of Figure 2 is not the same? What about the film thickness? The figure should be merged into one plot with the same legend.

      6.  The reason of the different error bar in Figure 3 should be discussed.

       7.  “Based on the previous experiments on the change of the optical power density, the optical power density is determined at 60 mW/cm2 when studying the influence of the UV irradiation time on the AgNWs” It is hard to understand the function of the sentence.

         8.  The welding effect under SEM is not conclusive compared to the widely reported results, e.g., Ref. 32. Better SEM images will be needed or improved treatment condition to be explored.

         9What is the UV treatment condition used in Figure 6? Why Figure 6 (a) and (b) have small particle precipitation, the weak welding is only visible in Figure 6 (a)? Also from the SEM image, the diameter of the AgNWs is more than 30 nm.

        10.  More detailed experimental information in the device application section is needed. Why 400-780 nm light is used  as UV welding, what is the relationship with the previous sections?

Author Response

Dear reviewer

Thank you very much for your email of Aguest 9, 2022 concerning our manuscript (nanomaterials-1846470 entitled “Performance enhancement of silver nanowire based transparent electrodes by ultraviolet irradiation"). We have modified the manuscript suitably following the concerns and suggestions received from the reviewer. In response to your letter, we would like to reply to the comments as detailed below:

Yours sincerely,

Mr. Shengyong Wang

This manuscript by Wang and co-workers reports a work about the influence of ultraviolet (UV) irradiation on the performance enhancement of silver nanowire (AgNW) based transparent electrodes upon transmittance, sheet resistance and the application on silicon photodetector. It is an interesting topic related to transparent electrodes upon their application. The result is interesting but inconclusive and short of novelty. It will be more suitable to publish in a special journal rather than Nanomaterials.

Here are only some comments:

1. The authors focus on the interplay between UV light density, wavelength, and irradiation time on AgNWs films as a three-dimensional picture. However, there are only few points used to  conclude the results. The most data are only related to 380 nm UV light within  6 mins illumination.  Different wavelengths may need variable illumination time or density to reach optimal effects, but no data are presented.

 Thank you for the reviewer’s suggestion, The ultraviolet welding principle of silver nanowire is that the absorption of ultraviolet light causes the temperature of the contact point between the wire and the wire to increase so as to complete the welding. Meanwhile the 380nm UV light is chosen as the research object because the highest absorption peak intensity of silver nanowires is at 380nm. This is also proved by the conclusion in Fig. 3 (c). Silver nanowires have the best welding effect because they absorb the most 380nm UV light. When the silver nanowire is irradiated at other ultraviolet wavelengths, the optical power density must be increased to achieve better welding effect, which will accelerate the precipitation of small particles on the surface of the silver nanowire, thus affecting the electrical properties of the film.

2. The whole manuscript only shows the relative sheet resistance, it will nor mean too much, it just indicates the influence of the UV light. The absolute sheet resistance value upon different illumination parameters is much more important than the relative one.

Thank you for the suggestion, the detailed expressions has been added in the revised manuscript.In the experimental process, it is found that there will be some errors between the films prepared with the same parameters, and the absolute sheet resistance will show a unclear rule, which will have a certain impact on the experimental conclusion. Therefore, in order to obtain a more referential rule, the relative sheet resistance is shown. Meanwhile, the optimum parameters of the film are also mentioned in the conclusion.

In order to avoid the error of sheet resistance affecting the experimental law, the relative sheet resistance value is adopted.

3. Figure 1 is only a sketch of the experiment, how to conclude “As can be seen from Figure 1, the AgNWs changed from lap to tighter welding state, which was also the main reason for the improvement of the electrical performance of the AgNWs”?

We apologize for the error, the description of this sentence lacks rigor. It has been modified in the text.

As can be seen in Figure 1, the overlapping regions of the silver nanowires will become more tightly connected when exposed to UV radiation.

4. More precise description of the experiment is needed. “Subsequently, for a period of 10 s, the samples were put at the homogenizer for 1 min, which operated at the speed of 1500 rpm”.

 we have proofread the manuscript again and corrected the errors.

In order to make the solution on the substrate evenly distributed, the homogenizer should be started after the solution on the substrate is stationary for 10 seconds. The speed of the homogenizer is 1500r/min, and the time is 1min.

5. Figure 2 should start from e.g., 350 nm, the most absorption feature of AgNWs should be included.  Why the transmittance at 2000 nm in the two parts of Figure 2 is not the same? What about the film thickness? The figure should be merged into one plot with the same legend.

Thanks to the reviewers for their suggestions on this paper. The difference in transmittance at 2000nm is due to the error caused by the change of the test instrument and has been corrected in the figure. Silver nanowire film is composed of multiple nanowire, which belongs to heterogeneous anisotropic film, which is very unfavorable to the test of film thickness. According to the principle of film formation, film thickness is related to diameter. When there is only one layer, the film thickness is about 30-35nm. The two images were combined and an absorption map of the silver nanowires under UV light was added. 

Please find the modified picture in the attachment.

Figure 2. Transmittance of AgNWs with different UV irradiation times (400-11000 nm). Insert of the figure shows the absorption of  AgNWs at300-500nm.

6. The reason of the different error bar in Figure 3 should be discussed.

Thank you for the suggestion, the detailed expressions has been added in the revised manuscript.

In Fig. 3(a), when the UV welding light power is too low, the influence on the film is relatively small, and the error bar is very small. However, with the gradual increase of the optical power, the welding effect and particle precipitation become more obvious, which also leads to the increase of the error bar. As shown in Fig. 3(b), with the extension of welding time, the error bar is large in the second minute because the welding is not completely completed, and the error bar is small when the welding is basically completed three minutes later. In Fig. 3(c), the error bar of the test results near the UV irradiation wavelength of 380nm is small, which is caused by the strong absorption capacity of silver nanowires to 380nm UV light, resulting in better welding results

7. “Based on the previous experiments on the change of the optical power density, the optical power density is determined at 60 mW/cm2 when studying the influence of the UV irradiation time on the AgNWs” It is hard to understand the function of the sentence.

Thank you for the reviewer’s suggestion. It is necessary to determine the light power density and wavelength of UV light source when studying the influence of UV irradiation time on the welding effect of Ag nanowire. In the first set of experiments, 60mw/cm2 was determined as the most suitable optical power density, so this parameter was adopted for the next set of experiments. According to the reviewer’s suggestion, we have modified the sentance in revised manuscript.

When studying the effect of UV irradiation time on silver nanowire film, the UV power density is fixed as 60mW /cm2.

8. The welding effect under SEM is not conclusive compared to the widely reported results, e.g., Ref. 32. Better SEM images will be needed or improved treatment condition to be explored.

Thank you for the suggestion,by analyzing the change of square resistance and SEM image of silver nanowire film, it is proved that ultraviolet irradiation can bring about successful welding effect. The welding method adopted in reference 32 is thermal welding. Thermal welding applied to the whole film can provide a better effect on the welding of silver nanowires, so a more obvious effect can be seen in the SEM image. However, when acting as a transparent upper electrode in the device, the excessive energy of hot welding will cause damage to other parts of the device. However, UV welding has little effect on the device because of its non-contact mode.

9. What is the UV treatment condition used in Figure 6? Why Figure 6 (a) and (b) have small particle precipitation, the weak welding is only visible in Figure 6 (a)? Also from the SEM image, the diameter of the AgNWs is more than 30 nm.

Thank you for the reviewer’s suggestion for this paper. The welding conditions used in Fig. 6 are 60mw /cm2, 380nm,3min. This is a complement to the previous microscopic morphology, and the precipitation of particles is due to the accelerated degradation of silver nanowires by UV irradiation. Figure 6(a) mainly shows the welding effect of silver nanowires in many places, and Figure (b) mainly shows the phenomenon of particle precipitation after ultraviolet light irradiation. The reason why the diameter of the silver nanowire is greater than 30nm in the figure is as follows: Some distortion is caused when the image is scaled after the scale is marked, and the diameter of the silver nanowire should be 30-35nm. Changes have been made in the text.

Figure 6 is a supplement to the previous microscopic morphological representation. Figure 6(a) mainly shows the welding effect of silver nanowires in many places, and Figure 6 (b) mainly shows the phenomenon of particle precipitation after ultraviolet light irradiation.

10. More detailed experimental information in the device application section is needed. Why 400-780 nm light is used  as UV welding, what is the relationship with the previous sections?

Thank you for the reviewer’s question, more experimental details have been added. We apologize for the mislabeling of the quantum efficiency test band and have corrected it in time. Since the main corresponding band of silicon is visible light, the quantum efficiency of the 400-780nm band is tested.

The structure of the device is Ag/Si/AgNWs/Ag, in which the upper and lower silver are ring electrodes for eliciting test signals, and the silver nanowire is coated on the upper surface of the whole silicon as a transparent electrode, and silicon is the active layer of the whole device. AgNW-contained solution was spin-coated on a Si substrate, which spontaneously forms a Schottky device[33]. Compared with uncoated silver nanowires, due to the good transparency of the AgNWS conductive layer, more photons can be driven into the photoactive Si material, thus enhancing the light response. Meanwhile, the contact between the welded silver nanowires and Si substrate is closer, which also reduces the sheet resistance of the silver nanowires, and more charge carriers are injected into the organic layer through the anode, thus improving the quantum efficiency of the device.

[33] Kim, H.-S.; Patel, M.; Kim, H.; Kim, J.-Y.; Kwon, M.-K.; Kim, J. Solution-Processed Transparent Conducting Ag Nanowires Layer for Photoelectric Device Applications. Materials Letters 2015, 160, 305–308, doi:10.1016/j.matlet.2015.07.142.

Round 2

Reviewer 2 Report

This revised manuscript by Wang and co-workers indicated the authors took some comments into consideration.

1. The focus of the manuscript is on the interplay between the UV light density, wavelength, and the irradiation time on AgNWs films to optimize the film with potential application as a transparent electrode, e.g., on silicon photodetector. To achieve the optimal condition, in Fig. 3(a), more curves with different UV exposure time, e.g., from 0 to 10 mins must be included, like in Figure 2. Only one curve cannot offer enough evidence to make a conclusion. The same should be applied to Figure 3 (b) upon different wavelength and (c) upon variable UV irradiation time.

2. “With the change of the irradiation time, the transmittance only fluctuates slightly, which may be caused by the slight difference between the spin-coated samples”. Based on the experiment, the starting point the pristine AgNWs film with 0 min illumination, followed by different UV irradiation time from 0 to 6 mins, the optical properties were measured after step. As the starting sample is the same, how the “between the spin-coated samples” is involved these data? If different samples are used, as there is no information of the thickness available, the transmittance can be changed a lot dependent upon AgNWs film thicknesses.

3. Lines 134-137: “The irradiation time is controlled at 1 min because the influence of the first minute of the irradiation on the AgNW film can be observed readily” There is no evidence to show in the manuscript to support the conclusion. Data of radiation time dependent sheet resistance are required here.

4. When the Figures of Merit (FOM) for transparent electrode is discussed to help us to judge how well the silver nanowire network behaves as a transparent conductive electrode, you cannot focus on only one FOM. Here is a good example, “Figures of Merit for High-Performance Transparent Electrodes Using Dip-Coated Silver Nanowire Networks, Journal of Nanomaterials,  Volume 2012, Article ID 286104, 7 pages, doi:10.1155/2012/286104”. The two key parameters of AgNWs film are sheet resistance and optical transmittance, which must be discussed together regarding to their relationship (Nanomaterials 2018, 8, 628; doi:10.3390/nano8080628). Only the relative sheet resistance Rs/Rs0 (Rs: real-time sheet resistance, Rs0: initial sheet resistance) is nonmeaningful. The absolute square resistance is the key. The author discussed the sheet resistance but using the change of Rs/Rs0. The data of the real-time sheet resistance Rs must be directly presented.

5. Lines 223-231, reference is need at least to confirm AgNW spin-coating on a Si substrate to spontaneously form a Schottky device.  “uncoated” should be “unwelded”.  “However, the contact between the welded silver nanowires and Si substrate is closer” can reduces the sheet resistance of the silver nanowires?

6. The paragraph between lines 238 and 247 has no real relationship with UV-welded AgNWs transparent electrode applied on devices,although there are plenty of investigation of UV effects on devices from materials degradation, passivation, etc. “ZNo-based” should be “ZnO-based”.

Author Response

Dear reviewer

Thank you very much for your email of Aguest 19, 2022 concerning our manuscript (nanomaterials-1846470 entitled “Performance enhancement of silver nanowire based transparent electrodes by ultraviolet irradiation"). We have modified the manuscript suitably following the concerns and suggestions received from the reviewer. In response to your letter, we would like to reply to the comments as detailed below:

Yours sincerely,

Mr. Shengyong Wang

1. The focus of the manuscript is on the interplay between the UV light density, wavelength, and the irradiation time on AgNWs films to optimize the film with potential application as a transparent electrode, e.g., on silicon photodetector. To achieve the optimal condition, in Fig. 3(a), more curves with different UV exposure time, e.g., from 0 to 10 mins must be included, like in Figure 2. Only one curve cannot offer enough evidence to make a conclusion. The same should be applied to Figure 3 (b) upon different wavelength and (c) upon variable UV irradiation time.

Thanks to the reviewers for their suggestions on this paper. In this paper, the influence of three controllable parameters on the UV welding performance of silver nanowires was studied. In the experiment, the control variable method is adopted, and the other parameter is changed when two parameters are selected to study the law of the influence of this parameter on UV welding. It can be seen from Figure 3 that the rule is quite obvious, and the reliability of the rule has been verified by repeated experiments during the experiment. In addition, it is compared with the law in the same research on ultraviolet welding to verify the credibility of the conclusion.

2. “With the change of the irradiation time, the transmittance only fluctuates slightly, which may be caused by the slight difference between the spin-coated samples”. Based on the experiment, the starting point the pristine AgNWs film with 0 min illumination, followed by different UV irradiation time from 0 to 6 mins, the optical properties were measured after step. As the starting sample is the same, how the “between the spin-coated samples” is involved these data? If different samples are used, as there is no information of the thickness available, the transmittance can be changed a lot dependent upon AgNWs film thicknesses.

We apologize for the error, the description of this sentence lacks rigor. It has been modified in the text. Multiple samples with different irradiation times were prepared because of concerns about damage in multiple tests with a single sample. The main factor affecting the transmittance of silver nanowire film is the concentration of the dispersion solution. Therefore, the concentration of dispersant and the number of rotating coatings used in the preparation of multiple samples are the same. Such preparation conditions make the thickness of each sample the same.  Irregular small differences in transmittance between different samples may be due to the preparation of the substrate used for the test.

With the change of irradiation time, the transmittance fluctuates only slightly irregularly, which may be caused by the subtle differences between the samples coated with spins and the substrate preparation process.

3. Lines 134-137: “The irradiation time is controlled at 1 min because the influence of the first minute of the irradiation on the AgNW film can be observed readily” There is no evidence to show in the manuscript to support the conclusion. Data of radiation time dependent sheet resistance are required here.

Thanks to the reviewers for their suggestions on this paper. This conclusion is obtained by referring to 23 “Facile and Efficient Welding of Silver Nanowires Based on UVA‐Induced Nanoscale Photothermal Process Roll‐to‐Roll Manufacturing of High‐Performance Transparent Films”. As shown in the figure below, the variation of sheet resistance in the first minute is most obvious for samples with different concentrations and diameters.

The figure can be viewed in the attachment.

4. When the Figures of Merit (FOM) for transparent electrode is discussed to help us to judge how well the silver nanowire network behaves as a transparent conductive electrode, you cannot focus on only one FOM. Here is a good example, “Figures of Merit for High-Performance Transparent Electrodes Using Dip-Coated Silver Nanowire Networks, Journal of Nanomaterials,  Volume 2012, Article ID 286104, 7 pages, doi:10.1155/2012/286104”. The two key parameters of AgNWs film are sheet resistance and optical transmittance, which must be discussed together regarding to their relationship (Nanomaterials 2018, 8, 628; doi:10.3390/nano8080628). Only the relative sheet resistance Rs/Rs0 (Rs: real-time sheet resistance, Rs0: initial sheet resistance) is nonmeaningful. The absolute square resistance is the key. The author discussed the sheet resistance but using the change of Rs/Rs0. The data of the real-time sheet resistance Rs must be directly presented.

Thank you for the reviewer’s suggestion. The real-time resistance R data are given.

Table 1 shows the variation of square resistance of silver nanowires before and after welding under the influence of different factors.

Table. 1 Variation of sheet resistance of AgNWs under ultraviolet irradiation under different influencing factors. 

influence factor

30mw

60mw

90mw

120mw

150mw

-

Optical power density

Before irradiation(Rs0)/Ω·sq-1

98.5928

114.323

107.0665

101.30134

112.1804

-

After irradiation(Rs)/Ω·sq-1

88.369775

84.4353

87.34624

129.2432

170.1116

-

Rs/RS0

0.8963

0.7386

0.8158

1.2758

1.5164

-

influence factor

1min

2min

3min

4min

5min

6min

irradiation time

Before irradiation(Rs0)/Ω·sq-1

114.323

126.213

128.354

122.7258

115.5719

124.06775

After irradiation(Rs)/Ω·sq-1

89.77218

77.87644

73.54312

72.39732

69.75925

72.6567

Rs/RS0

0.79905

0.6170

0.56596

0.59091

0.6036

0.57137

influence factor

350nm

360nm

370nm

380nm

390nm

400nm

ultraviolet wavelength

Before irradiation(Rs0)/Ω·sq-1

123.2068

114.678

113.164

128.354

109.01

117.949

After irradiation(Rs)/Ω·sq-1

88.8007

81.857

77.8225

73.54312

92.6678

99.147

Rs/RS0

0.7207

0.7138

0.6877

0.57297

0.85

0.8406

5. Lines 223-231, reference is need at least to confirm AgNW spin-coating on a Si substrate to spontaneously form a Schottky device.  “uncoated” should be “unwelded”.  “However, the contact between the welded silver nanowires and Si substrate is closer” can reduces the sheet resistance of the silver nanowires?

we have proofread the manuscript again and corrected the errors. In solution-processed transparent Tips nanowires layer for photoelectric device, applications, in order to inspect the interface of AgNW/Si, C-V characteristics were investigated by Mott-Schottky analyses. It is proved that silver nanowires can form Schottky junctions with silicon. The degree of binding to the substrate does not affect the sheet resistance of silver nanowires. The text has been corrected

Compared with unwelded silver nanowires, due to the good transparency of the AgNWS conductive layer, more photons can be driven into the photoactive Si material, thus enhancing the light response. However, the contact between the welded silver nanowires and Si substrate is closer, it provides more ways to transmit electrons. UV welding reduces the sheet resistance of the Ag nanowires, and more carriers are injected into the organic layer through the anode, thus improving the quantum efficiency of the device.

6. The paragraph between lines 238 and 247 has no real relationship with UV-welded AgNWs transparent electrode applied on devices,although there are plenty of investigation of UV effects on devices from materials degradation, passivation, etc. “ZNo-based” should be “ZnO-based”.

Thanks to the reviewers for their suggestions on this paper. The purpose of this analysis is to discuss whether UV welding technology can be applied to each device. The conclusion is that it needs to be used without affecting the rest of the functional layers. Errors in the text have been corrected.
